# LTβR Signaling Controls Lymphatic Migration of Immune Cells

**DOI:** 10.3390/cells10040747

**Published:** 2021-03-29

**Authors:** Wenji Piao, Vivek Kasinath, Vikas Saxena, Ram Lakhan, Jegan Iyyathurai, Jonathan S. Bromberg

**Affiliations:** 1Department of Surgery, University of Maryland School of Medicine, Baltimore, MD 21201, USA; wpiao@som.umaryland.edu (W.P.); rlakhan@som.umaryland.edu (R.L.); 2Center for Vascular and Inflammatory Diseases, University of Maryland School of Medicine, Baltimore, MD 21201, USA; VSaxena@som.umaryland.edu (V.S.); JIyyathurai@som.umaryland.edu (J.I.); 3Renal Division, Department of Medicine, Brigham and Women’s Hospital, Harvard Medical School, Boston, MA 02115, USA; vkasinath@bwh.harvard.edu; 4Department of Microbiology and Immunology, University of Maryland School of Medicine, Baltimore, MD 21201, USA

**Keywords:** lymphotoxin, lymphotoxin β receptor signaling, Treg migration, non-canonical nuclear factor κB pathway, lymphatic endothelial cells

## Abstract

The pleiotropic functions of lymphotoxin (LT)β receptor (LTβR) signaling are linked to the control of secondary lymphoid organ development and structural maintenance, inflammatory or autoimmune disorders, and carcinogenesis. Recently, LTβR signaling in endothelial cells has been revealed to regulate immune cell migration. Signaling through LTβR is comprised of both the canonical and non-canonical-nuclear factor κB (NF-κB) pathways, which induce chemokines, cytokines, and cell adhesion molecules. Here, we focus on the novel functions of LTβR signaling in lymphatic endothelial cells for migration of regulatory T cells (Tregs), and specific targeting of LTβR signaling for potential therapeutics in transplantation and cancer patient survival.

## 1. Introduction

The lymphatic network drains the peripheral tissues and returns interstitial fluid back into the circulatory system. The lymphatics facilitate immune cell migration from the periphery to draining lymph nodes (dLN). This migration plays crucial roles in immune surveillance, initiation of immunity, and induction of tolerance. As the primary route of the lymphatic system, lymphatic vessels and endothelial cells possess important immunomodulatory roles in addition to their transport functions. However, the molecular cues that regulate the entry of immune cells from peripheral, non-lymphoid tissues into afferent lymph vessels and, in particular, their subsequent migration from afferent lymphatics into the dLNs, remain elusive. Via direct interaction with immune cells, lymphatic endothelium or lymphatic endothelial cells (LECs) have been shown to modulate dendritic cell (DC) function, help maintain T cell homeostasis, and modulate T cell activation by transferring antigens and producing cytokines. Most importantly, LECs which comprise the afferent lymphatic vessels, orchestrate leukocyte trafficking from peripheral tissue to LNs by producing chemokines (including CCL19, CCL21, and CXCL12) and modulating an array of adhesion molecules (including ICAM-1, VCAM-1, and VE-Cadherin). LECs also produce sphingosine-1-phosphate (S1P), which promotes lymphocyte egress [1,2]. Several signaling receptors on LECs (including VEGFR-C [3], S1P receptors [4], CLEVER-1 [5], and LTβR [6,7] regulate these chemokines and adhesion molecules. Among these, lymphotoxin β receptor (LTβR) is highly expressed on LECs and constitutively signals through the non-canonical- NF-κB-inducing kinase (NIK) pathway. In this review, we will focus on the roles of LTβR signaling in endothelial cells on lymphatic trafficking of immune cells, especially of regulatory T cells (Tregs).

## 2. LTβR and Its Ligands

LTβR is a type 1 single transmembrane protein and member of the tumor necrosis factor receptor (TNFR) family that plays a critical role in the development of secondary lymphoid organs. An understanding of its importance to the generation and modulation of immune responses is still increasing steadily. The human LTβR gene is located on chromosome 12p13, in the same locus as two other members of the TNFR family—TNFR1 and CD27 [8]. The human LTβR gene shares 76% homology at the nucleic acid level with its mouse counterpart, located on chromosome 6 [9]. LTβR is expressed by a variety of immune cells, including lymphoid tissue stromal cells [10], myeloid cells [11], monocytes, alveolar macrophages in the lung [12], mast cells [13], DCs [14], and most adherent primary cells and tumor cell lines [10]. The protein is conspicuously absent in T cells, B cells, and NK cells [8].

Fully glycosylated LTβR is a 61 kDa protein that decreases to a theoretical mass of 47 kDa in the absence of glycosylation [8]. The cytoplasmic domain of the protein consists of 175 amino acids, including an area in proximity to the cell membrane with abundant proline residues, a feature that LTβR shares with other TNFR family proteins—CD40, CD30, herpes virus entry mediator (HVEM), and CD27—that interact directly with TNF receptor associated factor (TRAF) proteins [8].

LTβR binds strongly to two different natural ligands in homeostatic conditions (Figure 1), LTα1β2 and LIGHT (TNFSF14) [8]. In contrast to LTα1β2, which does not bind to any other molecule besides LTβR, LIGHT binds to herpes virus entry mediator (HVEM, TNFRSF14) as well [8]. Both membrane-bound and soluble forms of LIGHT bind to LTβR, whereas LTα1β2 is a purely membrane-bound protein, anchored by the LTβ subunit [15]. In addition, decoy cell receptor 3 (DcR3) is a secreted factor identified in several different malignancies, such as cancers of the lung, colon, GI tract, and brain, that binds to LIGHT and blocks interaction of LTβR and LIGHT [16]. Besides LTα1β2, LTα also exists in a secreted homotrimeric form (LTα3) that does not bind to LTβR and instead resembles TNF in its affinity for TNFR1 and TNFR2 [17]. Importantly, the membrane-bound ligand for LTβR, LTα1β2, is mostly expressed on T and B cells, which lack LTβR expression, indicating a unique role of LTα1β2-LTβR signaling in the communication between lymphocyte and LTβR-bearing cells.

## 3. Activation of LTβR Initiates the NFκB Signaling Pathways

Interaction of LTβR with LTα1β2 or LIGHT leads to a clustering of neighboring receptors, activating an intracellular signal transduction cascade. Binding of LTα1β2 to LTβR results in the swift shuttling of the TRAF proteins into the vicinity of cytoplasmic domain of the LTβR [18]. TRAF2, TRAF3, and TRAF5 are members of the TRAF family of zinc finger proteins, Really Interesting New Gene (RING), that bind directly to a small region (PEEGDPG) in the cytoplasmic portion of LTβR, though their exact sites of contact are not identical [8,19,20]. The shared structure of TRAF2, TRAF3, and TRAF5 resembles a mushroom; the receptor-binding, C-terminal domain forms the hood, and the N-terminal domain, which interacts with signaling molecules, forms the stalk [21]. The conserved residues in TRAF2, TRAF3, and TRAF5 that interact with LTβR cluster in three discrete locations in the C-terminal domains [21].

The NFκB family of transcription factors consists of five members: p65 (RelA), RelB, c-Rel, p50 (NFκB1), and p52 (NFκB2) [22]. Two distinct signaling cascades that result in NFκB-mediated transcription have been identified—the canonical and non-canonical pathways. The activation of the canonical NFκB signaling pathway is immediate and occurs within minutes, as it does not rely on novel gene expression like the non-canonical pathway does [8,23]. The canonical pathway is initiated classically by TNFR1, the activation of which does not require TRAF3 recruitment. Inhibitor of κB (IκB) sequesters the transcription factor RelA/p50 complex in the cytoplasm, covering its nuclear localization signal sequence [8]. Phosphorylation of IκB by IκB kinase (IKK) results in its degradation, liberating RelA/p50. Subsequently, RelA/p50 traffics to the nucleus, where it initiates gene expression of several pro-inflammatory effectors (Figure 2).

## 4. The Non-Canonical NFκB Signaling Pathway Is the Major Route for LTβR-Mediated Signal Transduction in LEC

The non-canonical NFκB signaling pathway transduced through LTβR is a true counterpart to the canonical pathway, as its effectors can attenuate the expression of genes that are governed by the canonical pathway [17]. However, the non-canonical NFκB signaling pathway requires a longer duration for signal transduction than the canonical pathway, and its effects are observed within a period of hours, since the pathway involves slow posttranslational modification processes. In resting cells, de novo synthesized NIK, a key component of the non-canonical NFκB pathway activation, is immediately bound and targeted by TRAF3-TRAF2-cIAP1/2 E3 ubiquitin ligase complex for K48-polyubiquitination and proteasomal degradation, to keep NIK expression low under normal conditions [24]. Within the complex, the cellular inhibitor of apoptosis (cIAP)1 and cIAP2 (cIAP1/2) which are brought by TRAF2, are responsible for the constitutive NIK degradation. TRAF3 has no direct E3 ubiquitin ligase activity towards NIK, and does not directly bind cIAP1/2. cIAP1/2 is also responsible for the inducible degradation of TRAF3 in response to non-canonical NFκB activation signals [23]. Under the non-canonical signaling, TRAF2 is recruited to the receptor complex, where it mediates K63 ubiquitination of cIAP1/2 and, thereby, stimulates the K48 ubiquitin ligase activity of cIAP1/2 towards TRAF3. The proteasomal degradation of TRAF3 leads to NIK stabilization and accumulation [25]. The question why cIAP1/2 is constitutively active towards NIK in resting cells and has to be activated by TRAF2 to ubiquitinate TRAF3 in signal-induced cells may be explained by the LTβR signaling in LECs, in which TRAF3 is constitutively bound to the receptor and has detectable NIK expression [6]. The receptor bound TRAFs seemed to be more susceptible for degradation, which might involve the rapid receptor complex internalization required for LTβR-non-classical NFκB signaling [26] (Figure 2).

Next, by a feedback loop, NIK acts with IKKα to be autophosphorylated prior to its phosphorylation of IKKα. Subsequently IKKα binds and phosphorylates serines 866 and 870 of p100, the NF-κB2 precursor protein [27], resulting in p100 ubiquitination by beta-TrCP ubiquitin ligase, and degradation by the 26S proteosome [28]. The p100 phosphorylation is dependent on protein synthesis. Processing of p100 by the 26S proteosome results in the formation of p52. P52 is the active transcription factor that associates with RelB and translocates to the nucleus, where it in turn promotes the transcription of p100 [29]. Therefore, the transcription factors involved in the canonical and non-canonical NFκB signaling pathway are distinct: RelA/p50 for the canonical pathway, and RelB/p52 for the non-canonical pathway.

TRAF3 acts as a negative regulator of the non-canonical pathway, as it may interact with the N-terminus of NIK at residues 78–84, labeling it for degradation and interfering with its processing of p100 [24]. On the other hand, TRAF2 activity appears to be essential for activation of both the canonical and non-canonical NFκB signaling pathways, as mouse fibroblasts that lack TRAF2 expression are unable to activate either pathway in response to LIGHT treatment [30]. This dual activity of TRAF2 is a key regulatory switch for the canonical and non-canonical pathways. Engagement of IKKγ by TRAF2 may lead to progression along the canonical pathway, whereas its interaction with IKKα may trigger activity of the non-canonical pathway. The activity of the various TRAF proteins may also vary by cell type. For example, both TRAF2 and TRAF3 appear to inhibit the activity of NIK in B cells by enabling the interaction of cIAP1 and cIAP2 with NIK, thereby promoting its degradation [23]. At baseline, TRAF3 is bound to LTβR in LECs, which display constitutive activation of the non-canonical NFκB signaling pathway. Sequestration of TRAF3 from the LTβR signaling complex using a permeable decoy peptide (nciLT) abolished the non-canonical branch of NFκB in LECs [6] (Figure 3).

## 5. Signaling through LTβR Results in the Expression of Chemokines, Pro-Inflammatory Cytokines, and Adhesion Molecules

Activation of the non-canonical NFκB pathway through LTβR-mediated signaling leads to the production of a variety of important chemokines and pro-inflammatory cytokines important in the mounting of an immune response, including CCL19, CCL21, CXCL3, CXCL12, and BAFF [31]. CCL19 and CCL21 are key chemokines implicated in the secondary lymphoid organ development, immune surveillance, and lymphocyte homing, and CXCL12 contributes to the development of early stage B cells [32,33,34,35]. The importance of LTα1β2-LTβR signaling to the development of the immune system is epitomized by the finding that LTα^−/−^, LTβ^−/−^, LTβR^−/−^ mice are devoid of LNs. Interestingly, mice deficient in NIK, IKKα, and RelB also lack lymph nodes [8]. LTβR-mediated signaling also augments the IKKα-independent expression of the pro-inflammatory cytokines CCL4 and CXCL2 in mouse embryonic fibroblasts (MEFs), indicating dependence on progression through the canonical NFκB signaling pathway, which is also required for induction of adhesion molecules and inflammatory chemokines/cytokines such as VCAM-1, MIP-1β, and MIP-2 in response to LTβR ligation [31]. Given the induction of leukocyte homing chemokines by LTα1β2-LTβR signaling, and the defects of lymphoid organ and tissue development in LT family member deficient mice, this suggests a critical role for LTβR signaling in immune cell lymphatic trafficking. Surprisingly, this role has not been well appreciated, and the mechanisms controlling lymphatic migration of immune cells remain poorly understood. Recently, our group recognized that peripheral dermal vascular endothelium and lymphatic endothelial cells (LECs) expressed high levels of LTβR [6]. The LTβR in these cells signals predominantly via the constitutive and ligand-driven non-classical NFκB-NIK pathway to upregulate cell trafficking chemokines, hence regulating lymphatic transendothelial migration (TEM).

## 6. Effects of LTβR Activation in Endothelial Cells

Mounting evidence highlights the critical influence of LTβR in determining the activity of blood vascular endothelial cells (BECs). In these cells, LTβR ligation activates both the canonical and non-canonical NFκB signaling pathways [36]. However, its stimulation of the canonical pathway is significantly weaker than the signaling through TNF-induced activation [36]. Activation of the canonical pathway in BECs by LTβR results in the expression of several pro-inflammatory genes, including E-selectin, VCAM-1, ICAM-1, and CXCL12 [36]. In fact, LTβR ligation results in the adherence of T cells to human umbilical vein endothelial cells (HUVECs) in vitro [36]. Activation of the non-canonical signaling pathway results in production of CXCL12 in HUVECs [36], and CXCL13 and CCL21 in LEC [37,38], and these chemokines are required for LN homing of B cells and CXCR5^+^ DCs [38]. However, these findings have not been linked to the immune cell lymphatic transendothelial migration (TEM).

The specific role of LTβR in LECs for immune cell migration has received little attention thus far. Recent findings demonstrate that both LTβR ligands, LTα1β2 [6,7] and LIGHT [39], signal through LEC LTβR to regulate Treg and DC migration, respectively. During inflammation, LIGHT surface expression was upregulated in Langerhans cells (LCs), and soluble LIGHT was increased in the skin after LPS stimulation, which also induces CCL19 and CCL21 in LECs. LTα1β2 expression on DC was less pronounced. LIGHT-deficient LCs had impaired efferent lymphatic migration to dLNs. Endothelial-specific LTβR deficiency also impaired DC migration, suggesting the important role of LIGHT-LTβR signaling on DC migration [39]. Blockade of LTβR-noncanonical NFκB-NIK signaling in LEC inhibited bone marrow derived DC lymphatic TEM in vitro and in vivo [6]. However, the inhibition is likely not caused by DC LTα1β2 induced LEC LTβR signaling, but by LTα1β2-high expressing Tregs, which interact with LTβR on afferent LECs, leading to activation of the non-canonical NFκB signaling pathway causing LEC structural changes. This change in morphology is associated with increased transmigration of Tregs across the endothelium [7]. Studies of LTβR signaling on LEC revealed predominant regulation of cell migrating related cell adhesion molecules, intercellular junction proteins, and most importantly, the cell migrating cytokines or chemokines [6,39], supporting the important roles for LTβR signaling for immune cell transendothelial migration (TEM). In the next sections, we will discuss the newly defined roles of LTα1β2-mediated LTβR signaling on immune cell lymphatic TEM.

## 7. LT Expression and Regulation

The heterotrimer of LTα1β2 is the major ligand of LTβR. Expression of LTα1β2 is restricted to cells of lymphoid lineage including T, B, natural killer (NK), and lymphoid tissue-inducer cells [40]. LTβ is exclusively anchored in the membrane and binding LTα to form membrane-anchored heterotrimers: LTα1β2 and LTα2β1 [41,42]. Unlike LTβ, LTα can also be secreted as a soluble LTα3 homotrimer. LTα1β2 mediates LTβR signaling, whereas LTα2β1 is a rare form of LT expressed by less than 2% of T cells and with an undefined biological role [43]. LTβR has an additional ligand, LIGHT (TNFSF14), which also interacts with herpesvirus entry mediator (HVEM) [17], however, unlike LTαβ and LTβR, whose deficiencies abolish lymphoid organ and tissue development and lymphoid microarchitecture disorder [44,45], LIGHT deficiency causes no such alterations [46]. LTβR is not present on lymphocytes, but is strictly expressed on nonhematopoietic endothelial [6,47], parenchymal, and stromal cells, and also on myeloid cells [48]. The restricted expression patterns of LT and LTβR enable directed cell–cell communication between lymphocytes and endothelial or stromal cells, thereby influencing various biological processes [49,50,51].

LTα1β2 is expressed on activated T or B cells, while naïve CD4 T cells express little to no LTα1β2. Although certain cytokines or other ligands have been mentioned to regulate LTα1β2 expression in different immune cell subsets [17], functionally relevant inducers have not been clearly identified. LTα1β2 upregulation in Jurkat T cells has been induced by protein kinase C-mediated Ets (E26 transformation-specific), NF-κB(p65/Rel), and Egr-1 (early growth response protein 1)/Sp1 (specific protein 1) promoter activation [52,53]. Whether these also occur in antigen specific TCR activated T cells is uncertain. Recently, our group identified that IL-2R signaling is the major route to enhance Treg LTα1β2 expression in T cells [51].

### 7.1. LTα1β2 in Tregs during Homeostasis

LTα1β2 is preferentially expressed and used by regulatory T cells (Tregs) for afferent lymphatic migration [7]. Tregs constitutively express the interleukin 2 receptor α chain (IL-2Rα; CD25) and rely on IL-2 for Foxp3 induction and Treg differentiation and maintenance [54]. IL-2 led to a dose-dependent increase of LTα1β2 on in vitro stimulated induced Tregs (iTregs), whereas activation by anti-CD3 TCR ligation caused a modest LTα1β2 increase, which was further enhanced by anti-CD28 mAb co-stimulation or by stimulation together with IL-2. Notably, blocking IL-2R with anti-CD25 blocking mAb markedly diminished anti-CD3-activated LTα1β2 expression [51], suggesting TCR/CD3-mediated LTα1β2 expression is indirectly induced by TCR/CD3-triggered IL-2 secretion. Analysis of various other CD4 T cell subsets indicated that Tregs, especially the activated iTregs, expressed the highest levels of LTα1β2. Freshly isolated unstimulated-thymus-derived natural Tregs (nTregs) expressed less LTα1β2 than iTregs; however, anti-TCR/CD3-activated nTreg expressed comparably high levels of CD25, Foxp3, and LTα1β2 as iTregs. LTα1β2 was highly expressed on the CD25^high^Foxp3^high^ iTreg fraction, at an intermediate level on CD25^int^Foxp3^int^, and minimally expressed on CD25^low^Foxp3^−^ non-Treg CD4 T cells. Notably, LTα1β2 was positively regulated by PI3K/Akt signaling, and LTα1β2 expression was not correlated directly with Foxp3 or CD25 expression. LTα1β2 expression in Tregs is promoted by IL-2R activation through NF-κB and MAPK pathways, since blocking either activity abolished IL-2-mediated LTα1β2 increases [51]. Thus, in homeostatic conditions, with constitutive expression of surface IL-2Rα (CD25), patrolling nTregs express modest levels of LTα1β2, allowing them to interact with LTβR-expressing lymphatic endothelium for transendothelial migration and maintain immune surveillance.

### 7.2. LTα1β2 Expression on Tregs during Inflammation

Toll-like receptors (TLRs) are major innate immune sensors that sense pathogen-associated molecular patterns (PAMPs). TLR2 forms heterodimers with TLR1 or TLR6 to recognize a broad spectrum of microbial and endogenous products. Toll-like receptor 2 (TLR2) is the only functional TLR expressed on TCR-primed or activated CD4 or CD8 T cells. Naïve CD4 has no TLR2 expression [55]. TLR2 signaling affects Treg expansion and function [56,57,58]. Treg subsets expressed much higher levels of TLR2 compared to non-Treg CD4 T cell subsets [51,56]. TLR2 co-stimulation by the TLR1/TLR2 agonist Pam3Cys-SK4 (P3C) or the TLR2/TLR6 agonist Pam2CSK4 (P2C) together with IL-2 further increased the expression of LTα1β2 on iTregs, but not on nTregs. In contrast, naïve or activated CD4 T cells expressed very little LTα1β2, and expression was unaffected by IL-2 or TLR2 co- stimulation. Similarly, activated human Tregs, but not activated non-Treg CD4 effector T cells, expressed high levels of TLR2, and TLR2 activation increased LTα1β2 expression on human Tregs, but not on effector T cells [51]. The differential responses to the TLR2 ligands among the T cell subsets was due to differences in the TLR2 expression, further proved by the use of TLR knock out cells [51]. Notably, other TLR ligands, such as lipopolysaccharides (LPS) (TLR4 agonist) and R848 (TLR7 agonist), had no effect on IL-2-stimulated LTα1β2 expression, indicating an important role for TLR2 on LTα1β2 expression in Tregs. TLR2 augmented classical NF-κB, ERK, and PI3K-Akt (Thr308), but not JNK signaling. TLR2 together with the IL-2R signaling increase LTα1β2 expression in the Treg subsets [51]. Thus, the specific regulation by TLR2 of LTβR ligand expression on activated Tregs indicates unique functions of mobilized Tregs during inflammation.

## 8. LTαβ-LTβR Signaling Regulates Immune Cell Lymphatic Migration

The major role of LTβR signaling on immune cell migration was first implied by its ability to activate endothelial and stromal cells to express cell adhesion molecules (V-CAM, I-CAM, and MAdCAM) [47,59,60], and to regulate expression of homeostatic chemokines (CXCL13, CCL19, CCL21) [61,62,63] required for trafficking and positioning of T and B cells to their respective niches in second lymphoid organs (SLO). Lymphocyte-epithelial crosstalk has also been studied for thymocyte egress. Functionally, the absence of LTβR signaling leads to the retention of mature T cells in the thymic medulla [64]. Given that activated LTβR in endothelium signals to regulate the expression of immune cell trafficking chemokines and cell adhesion molecules, we and others have been investigating the crosstalk between the blood or lymphatic endothelium and the leukocytes expressing high levels of LTβR ligands.

Leukocyte lymphatic migration initiates and modulates immune response and resolves inflammation. Migration of immune cells from the tissues to dLN uses afferent lymphatics as conduits. This migration is a tightly regulated multi-step process involving intravasation to the peripheral tissue lymphatics, intramural crawling leading to their propulsion to the dLN, guided by CCL21 dependent chemotaxis [65]. Mechanisms of afferent lymphatic leukocyte migration are cell specific, as T cells employ different molecules compared to those used by DC, neutrophils, or monocytes. We and others identified that naïve or activated CD4 T cell lymphatic migration is integrin-independent and employs the S1P-S1PR1/S1PR4 pathway [4,66]. Further Tregs but not non-Tregs use LTα1β2-LTβR signaling for afferent lymphatics migration [6,7]. This LT-dependent mechanism employed by Tregs is unique, and is not required to enter LN via the HEV, nor egress from the LN to efferent lymphatics [7].

### 8.1. Treg Migration

Treg trafficking between inflamed tissues and dLNs is essential for optimal immune suppression [67]. The migration of Treg from allografts to LN via afferent lymphatics is critical for graft survival, and cannot be supplanted by Treg migration from blood through HEV into the same LNs [68]. Distinct from non-Treg CD4 T cells, Treg specifically employ several molecular mechanisms to migrate through afferent lymphatics, to balance immunity and inflammation in both homeostatic and inflammatory condition.

#### 8.1.1. Homeostatic nTreg Migration

In the steady state, thymic-derived nTregs patrol the immune system to maintain immunological tolerance. Their path takes them from blood through tissues into lymph and back to blood. Once in the periphery, chemokine-driven homing via CCR7/CCL19 or CCL21 enable nTregs to migrate from blood to LNs [69,70]. CCR7 may also be required for nTreg migration from tissues to afferent lymphatics and lymphoid organs [71], although there are conflicting reports [72,73]. T-bet is also required for nTreg migration into afferent lymphatics and dLNs [74], implicating multiple mechanisms for this process.

Compared to naïve CD4 T cells which have almost no LT expression, nTregs express higher levels of membrane anchored LTα1β2, suggesting that nTregs have priority for interacting with the constitutively active LTβR on LEC. The nTregLTα1β2 -LEC LTβR interaction upregulates LEC chemokines CCL19/CCL21 secretion, and triggers LEC cell adhesion and intercellular molecule changes to facilitate nTregs transmigration [7,51]. This may account for the observation that chemokines like CCL19 and CCL21are constitutively expressed and control cell movement during homeostasis [75]. Human naïve Tregs are also equipped with high levels of LTα1β2 and modulate CCL21, VCAM-1, and VE-cadherin expression in vitro of cultured human LECs [51]. In in vitro transmigration assays, prevention of the LEC LTβR ligand-binding with LTβR-Ig (a pan LTβR blocker which LTβR Fc fused with mouse IgG1) and blockade of LEC LTβR-NIK signaling with a NIK inhibitor, each inhibited nTreg TEM toward CCL19. In an in vivo model of the lymphatic migration from footpad to popliteal dLNs, *Lta*^−/−^ nTreg lacking both LTα3 and LTα1β2 demonstrated significant decreases in migration to the dLNs. Similarly, LTβR-Ig pre-treated wild type nTregs were also inhibited from migration. Migration of naive non-Treg CD4 or CD8 T cells to the popliteal dLNs was unaffected by LTβR-Ig or genetic ablation of *Lta* [7]. In the ear pinna model of lymphatic migration, T cells encountering lymphatics were visualized by microscopy. LTβR-Ig pretreated nTreg and *Lta*^−/−^nTreg were impaired in their migration to Lyve-1^+^lymphatics and were mostly positioned outside of the lymphatic vessel lumen. In contrast, non-Treg CD4 T cells were not affected by LTβR-Ig. Wild type nTregs transferred into *Ltbr*^−/−^ also demonstrated impaired migration to the lymphatic vessel walls and lumens [7]. Collectively, the observations indicate that nTreg, but not non-Treg CD4 T cell, migration into lymphatic vessels depends on LTα1β2 interactions with LTβR on stromal or lymphatic endothelial cells.

Notably, LT-dependent migration mechanisms did not regulate Treg LN egress [7]. It would be interesting to know why the LT system affect only the nTreg afferent but not the efferent lymphatic migration. One answer might be due to different signaling pathways active in skin or interstitial tissue LECs compared to LN LECs. For example, we could not detect LTβR-mediated upregulation of CCL19/CCL21 or cell adhesion molecules in LN LECs treated with an anti-LTβR agonistic mAb (unpublished data).

#### 8.1.2. Activated Treg Migration during Infection and Inflammation

A wide-array of chemokine receptor expression enables Tregs to efficiently access inflamed tissues [76]. During inflammation, Treg migration from peripheral tissues to dLNs is accelerated compared to the steady state. The migration of CD44^high^ CD62L^low^ effector/memory Tregs to dLNs is doubled compared to non-Tregs, indicating Tregs are rapidly activated and recruited to sites of inflammation [77].

Activated nTregs and peripheral induced Tregs [78] with effector-like phenotype (CD44^high^CD25^high^CD62^low^) possess the highest levels of membrane LTα1β2 compared to naïve CD4 and activated non-Treg CD4 T cells. This allows them to interact with lymphatic endothelium more actively, stimulating LEC LTβR-canonical NFκB to upregulate VCAM-1 and LEC LTβR-NFκB-NIK to increase CCL21 and decrease VE-cadherin [51]. Activated *Lta*^−/−^ Tregs have no such effect. Thus, by modulation of LEC chemokine expression and tight junction structures, activated Tregs use LTα1β2-LTβR signaling axis to promote their TEM.

Notably, activated Tregs express high levels of surface TLR2, a major receptor for multiple pathogens, including bacteria, viruses, fungi, and parasites, and a receptor for endogenous ligands signifying tissue injury. During inflammation, activated effector T cells produced IL-2, stimulating IL-2R-mediated LTα1β2 increases on activated Tregs. Importantly, TLR2 signaling dramatically promotes IL-2R induced LTα1β2 on activated Tregs. Enhanced LTα1β2 expression promotes Treg lymphatic migration. Activation of other TLRs such TLR4 or TLR7 had no such effect, showing the specific role for TLR2 signaling on immune cell migration [51]. Using an in vivo model of islet transplantation, we observed that pancreatic islets which harbor endogenous TLR2 ligands, such as high mobility group box 1 (HMGB1) and hyaluronan (HA), promoted the migration of wild type, but not *Tlr2*^−/−^ Tregs from the graft to the dLNs, and prolonged allograft survival, due to enhanced LTα1β2 expression triggered by the endogenous TLR2 ligands in the islets [51].

### 8.2. Tregs License the LEC for Other Immune Cell TEM during Inflammation

LT-dependent regulation of LEC surface molecules induced endothelium structural changes, and LT-mediated chemokine induction [51]. LTα1β2-LTβR signaling endows Tregs with migratory advantage and allows Tregs to ensure the lymphatic TEM of other immune cells. Under noninflammatory conditions, nTregs equipped with high levels of LTα1β2 patrol tissues and tissue lymphatic vessels may maintain LEC LTβR constitutive activation. This activity may permit regulatory or effector T cell homing and recirculation to maintain immune surveillance of tissue-derived endogenous or exogenous antigens for suppression of immune responses by Treg and induction of immune responses by effector T cells. During acute inflammation, where TLR ligands and IL-2 are present, activated Tregs express the highest levels of LTα1β2 and are rapidly mobilized to cross lymphatic endothelium and facilitate a variety of leukocyte subsets crossing LECs into dLNs [51] (Figure 4). Enhanced LT-dependent lymphatic TEM may allow rapid antigen presentation between effector T cells and DCs in the dLNs to mount efficient immune responses. Enhanced LT-mediated TEM of effector cells out of inflamed site may also resolve inflammation and decrease local tissue destruction. LT-dependent Treg conditioning of lymphatic endothelial TEM gating might be required at multiple levels of the immune response to regulate systemic trafficking, suggests that this single Treg function might be able to regulate a multitude of cell types and their functions.

Due to the instability and fragility of Tregs in various conditions, and the complexity of the interactions of Tregs and endothelial cells, many other important molecules are undoubtedly involved in these processes. Particularly under inflammation, Tregs upregulate multiple checkpoint proteins which may vary under the influence of different inflammatory conditions. Further investigation of molecular mechanisms of Treg and effector T cell lymphatic migration are needed to develop sufficient therapeutic strategies.

## 9. Targeting LTαβ-LTβR Signaling to Regulate Immune Cell Migration

Multiple strategies have been developed to dampen or boost LTβR signaling in various cell types to attenuate autoimmune diseases/inflammation or cancer immunity, respectively. Targeting LTβR extracellular ectodomain with a soluble decoy receptor (LTβR-Ig), which competes with LTβR ligand binding, or with anti-LTβR antibody has been used to inhibit LTβR signaling [79,80]. For boosting LTβR signaling, cholesterol sequestration or depletion from the plasma membrane reportedly prevented LTβR internalization and then activated LTβR-canonical NFκB signaling in lung epithelia to promote pro-inflammatory responses against cancer cells [81].

Recently, our group employed an antagonistic peptide blocking approach by selectively targeting the canonical or non-canonical branches of LTβR-NFκB signaling to promote the desired immune cell migration and to suppress the unwanted inflammatory response, hence protecting transplanted islet allografts. The precise peptide-targeting approach could be a potential cancer therapy, especially since many cancers have persistently active LTβR signaling [6].

### 9.1. Targeting the Extracellular Ectodomain of LTβR

LTβR-Ig, a soluble decoy fusion protein comprised of the ectodomain of LTβR and Fc of human or mouse immunoglobulin G (Ig), binds not only to membrane bound LTα1β2 on T cells, but also to LIGHT, a costimulatory molecule expressed on T cells and DCs. As a receptor decoy, LTβR-Ig competes with and blocks LTβR binding to LTα1β2 and LIGHT. LTβR-Ig has been shown to alleviate various autoimmune diseases in mouse models of rheumatoid arthritis, colitis, experimental autoimmune encephalomyelitis (EAE), and late stage of type 1 diabetes [82,83,84,85]. However, these results have not been consistent across disease models [83]. It is hard to prove that LTα1β2 -LTβR signaling is essential for the development of certain autoimmune diseases, since LIGHT-mediated LTβR signaling plays a similar role in some T cell-mediated autoimmune diseases. As a pan-LTβR signaling inhibitor, LTβR-Ig blocks classical NFκB and non-classical NFκB-NIK pathways. Thus, mice treated with LTβR-Ig exhibit reduced chemokines and adhesion molecules as well as a reduction in the cellularity in the lymph nodes [86,87,88]. Work from our group showed that treatment with LTβR-Ig failed to improve cardiac allograft survival [89], while LT expression by Treg was required to prolong islet allograft survival [7]. Thus, specific targeting of single ligands or either arm of LTβR-NFκB signaling, may selectively control the immune cell migrations or inflammatory responses, and identify the precise pathway responsible for disease pathology.

Targeting the extracellular ectodomain of LTβR with agonistic LTβR mAbs was also employed to activate LTβR signaling to suppress tumor growth and enhance tumor chemosensitivity [80]. However, some cancer cells were refractory to this anti-LTβR mAb treatment.

### 9.2. Targeting the Intracellular Domain of LTβR to Block Non-Canonical NFκB-NIK Signaling

LTβR-NFκB-NIK constitutively signals in LEC to maintain local cell-homing chemokine levels, highlighting lymphatic vessel function as gatekeeper of LT-dependent immune cell recirculation. Under inflammation, ligand driven LTβR activation also signals through canonical NFκB to induce inflammatory chemokine/cytokines. Therefore, selective targeting of LTβR-NIK signaling, without increasing inflammatory responses, could potentially inhibit pathological immune cell trafficking.

The differential recruitment and binding of adaptor proteins TRAFs 2, 3, and 5 to the intracellular domain of LTβR initiates classical or non-classical NFκB signaling pathways. Mutagenesis studies indicate that TRAF2 and TRAF3 binding to different motifs in the intracellular signaling domains of LTβR bifurcates the two arms of NFκB signaling [26]. TRAF3 recruitment has been proposed as a hallmark of the NIK pathway [90,91]. Therefore, selective targeting TRAF2 or TRAF3 binding sites could separately block each arm of LTβR-NFκB signaling pathways. In LEC, TRAF3 is constitutively bound to the LTβR, which prevents NIK degradation and causes basal level constitutive NIK activation [6]. To regulate separate LTβR signaling pathways, we created decoy peptides comprised of the N-terminal cell-penetrating sequence of the *Drosophila* antennapedia peptide (RQIKIWFQNRRMKWKK) plus TRAF-binding motifs in LTβR to specifically target each arm of the NFκB pathway [6]. The antagonistic peptide targeting non-canonical NFκB-NIK signaling, nciLT (RQIKIWFQNRRMKWKKTGNIYIYNGPVL), harbored the sequence required for TRAF2 and TRAF3 recruitment and binding to the activated LTβR complex. nciLT specifically sequestered TRAF3 from the receptor complex, and did not prevent TRAF2 degradation, but did prevent NIK accumulation for non-canonical NFκB signaling (Figure 3). Thus, nciLT blocked LTβR-NIK activation induced RelB nuclear translocation, and specifically inhibited the late response genes of the non-canonical pathway (chemokines CCL21, CXCL12) without inhibition of LTβR-canonical NFκB signaling induced inflammatory cytokine/chemokine or cell adhesion molecule expression. In addition, NIK pathway inhibition enhanced T cell binding to LEC through integrin β4 and VCAM [6].

In the contact hypersensitivity (CHS) mouse model, hapten sensitization induces dermal DC migration to dLNs for T cell priming. Hapten challenge further induces primed T cell activation and effector T cell recruitment to the site of challenge. Administration of nciLT to the abdomen before sensitization, inhibited CHS 24 hours after challenge, with fewer T cells and DC infiltrating the ear compared with controls. Additionally, CCL21 production by LEC has a critical role for DC afferent lymphatic migration in CHS and was suppressed by nciLT. Inhibition by nciLT of DC migration to the dLNs likely reduced T cell priming. Treatment with nciLT in the ear pinna at the time of hapten challenge and elicitation inhibited CHS with reduced T cell infiltration. Thus, nciLT at this stage may have again interfered with DC migration to the dLN and also stimulation of primed T cells. Administering the peptides at the start of the resolution stage, nciLT sustained CHS, likely by preventing egress of the inflammatory infiltrate, thus causing the observed increase in T cell and DC infiltrates in the ear. In contrast, ciLT enhanced the resolution, possibly by inhibiting cytokines and promoting the egress of the inflammatory cells, as fewer CD3 and CD11c cells were present in the ear [6].

### 9.3. Targeting the Intracellular Domain of LTβR to Block Canonical NFκB Signaling

LTβR activation induces rapid TRAF2 recruitment to the LTβR receptor to initiate canonical NFκB. The antagonistic peptide targeting canonical NFκB-NIK signaling, ciLT (RQIKIWFQNRRMKWKKTPEEGAPGP) includes the (P/S/A/T) X(Q/E)E TRAF-binding motif required for TRAF2, but not TRAF3 binding to LTβR [6]. ciLT blocks both TRAF2 and TRAF3 recruitment to the LTβR complex. Thus, nuclear translocation of RelA and early response genes of the canonical pathway induced by LTβR activation are specifically inhibited [6]. TNFRI activation also recruits TRAF2 to the receptor complex to initiate classical NFκB signaling pathway. However, TNF-induced IKKα/β phosphorylation in LEC was not affected by ciLT pretreatment, indicating the blocking peptide is highly specific for LTβR signaling.

In the CHS model, administering ciLT in the ear pinna at the time of hapten challenge and elicitation also inhibited CHS with reduced T cell infiltration. ciLT blocked expression of classical NFκB-driven inflammatory chemokine (e.g., CCL2) and adhesion molecules (e.g., VCAM-1, ICAM-1) by LEC, which are upregulated and important during CHS. In addition, ciLT may have also enhanced egress of inflammatory cells out of the ear [6].

In vivo targeting of immune cell migration in CHS by the LTβR specific blocking peptides demonstrated LTβR relevant scenarios for disease development, prevention or resolution; and individual kinetic components of immune responses such as sensitization, elicitation, and resolution can be precisely targeted. Both lymphocytes and myeloid cells are influenced by these pathways. These findings open up the possibility for other applications and investigations of how signaling and trafficking through lymphatics regulate immunity. These peptides may serve as a foundation for compound screening and drug discovery for novel therapeutics to regulate immune responses.

### 9.4. Selective Targeting Confirms the Competitive Signaling Arms of LTβR

It is noteworthy that each arm of LTβR signaling competes against the other in LEC. Blocking the non-canonical NFκB pathway enhanced canonical NFκB pathway gene VCAM-1 expression; conversely, blocking the canonical NFκB pathway increased the non-canonical NFκB gene CCL21 production [6]. Molecular interactions indicated receptor downstream crosstalk between the two arms of the LTβR–NFκB pathways, via TRAF3 or TRAF2 recruitment to the receptor. TRAF2 is a positive mediator of canonical NFκB activation while serving as a negative regulator of the non-canonical pathway in CD40 signaling in B cells [92]. These contrasting roles of TRAF2 may also translate to its dual functionality in LTβR signaling in LEC. The competing roles of the two pathways also explain the conflicting regulation of downstream molecules. Embryonic fibroblasts from *aly/aly* mice, which express a mutated NIK, make VCAM-1 mRNA but fail to express surface VCAM-1 in response to LTβR stimulation [93]. This might be due to the abnormal NIK signaling disrupted VCAM-1 expression driven by canonical NFκB signaling, re-emphasizing the crosstalk between the two pathways.

## 10. Summary

Although LTβR signals to both canonical and non-canonical ΝFκΒ pathways, in LECs it predominantly signals by both a constitutive and ligand-driven non-canonical NFκB-NIK pathway. Constitutive NIK activation in LECs is required for leukocyte TEM and implicates its importance in lymphatic recirculation for all immune cells. LTα1β2 ligand-driven LTβR-NIK signaling on LECs triggers lymphatic endothelial structural changes and upregulation of migration receptors and chemokines, such as CCL21 or CXCL12, resulting in enhanced inflammatory leukocyte lymphatic migration out of tissues. Importantly, during infection or inflammation, the microbial pathogen-recognizing TLR2 directly enhances the mobilization of LT-dependent Treg lymphatic migration. The transmigrating activated Tregs license the lymphatic endothelium for the other immune cell egress out of the inflamed tissues. However, Treg homing into LNs via HEV blood endothelium is not LT-dependent, relying instead on surface integrins. As LTβR is also expressed on blood endothelial cells, FRC, lung epithelial cells, and most tumor cells, it will be important to determine whether separate LTβR signaling play similar or contrasting roles in these various cell types for migration. Thus, many questions still remain unanswered. Why do other cell types, such as naive T cells, activated effector CD4, CD8 T cells, B cells, and DCs not require LT for lymphatic migration? Do Tregs license LN egress via efferent lymphatic migration for other leukocytes? The peptides which selectively inhibit each arm of LTβR signaling may help us explore mechanisms underlying the complex signaling network, and serve as precision medicine against autoimmune diseases and cancer.

## Figures and Tables

**Figure 1 cells-10-00747-f001:**
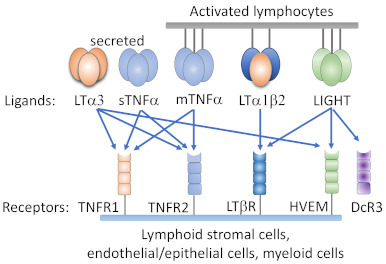
Ligands and receptors of the tumor necrosis factor (TNF)/lymphotoxin system. Both heterotrimeric LTα1/β2 and homotrimeric LIGHT bind to LTβR. Herpes virus entry mediator (HVEM) binds to membrane LIGHT and soluble homotrimeric LTα3. Membrane TNFα (mTNFα), soluble TNFα (sTNFα), and LTα3 bind to TNFRI and TNFR II.

**Figure 2 cells-10-00747-f002:**
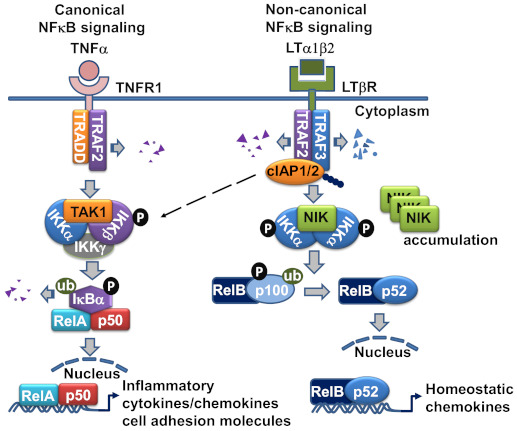
The LTβR signaling pathway in lymphatic endothelial cell (LEC). LTα1β2 engagement of LTβR initiates the recruitment of TRAF2 and TRAF3 to the LTβR complex, where TRAF2 and TRAF3 are degraded by cIAP1/2, and result in NF-κB-inducing kinase (NIK) stabilization and accumulation. NIK complexed with IKKα is activated and leads to the homodimeric IKKα phosphorylation. Eventually, the p100 precursor binding with RelB is cleaved to p52 and causes RelB-p52 heterodimeric complex translocation to the nucleus to initiate chemokine gene transcription. LTβR ligation also activates IKKα/β phosphorylation and RelA/p50 nuclear translocation, which leads to gene transcription of inflammatory and cell adhesion molecules. TRAF-2-mediated K63 ubiquitination of cIAP1/2 is also linked to the activation of canonical NFκB pathway.

**Figure 3 cells-10-00747-f003:**
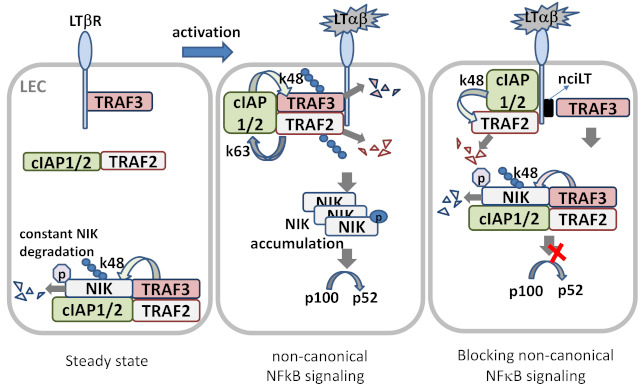
LTβR signaling in LEC. In steady state, newly synthesized NIK is rapidly bound by TRAF3 and targeted to the TRAF-cIAPs E3 ubiquitin ligase complex for K48-polyubiquitination and proteasomal degradation, where TRAF2 bridges TRAF3 and cIAPs. Low level NIK is unable to process p100 under normal conditions. In ligand (LTαβ or LIGHT)-stimulated LECs, the TRAF-cIAPs complex is recruited to the LTβR where cIAP1/2 is activated by TRAF2-mediated K63 ubiquitination, and the activated cIAP1/2 then targets TRAF3 for K48 ubiquitination and degradation. With the lack of TRAF3, de novo synthesized NIK accumulates and is activated via trans-phosphorylation. NIK then activates IKKα, leading to p100 processing and nuclear translocation of RelB/p52. Masking the TRAF3-binding site of LTβR by permeable blocking peptide nciLT leads to TRAF3 targeting NIK for degradation, and hence p100 processing is blocked.

**Figure 4 cells-10-00747-f004:**
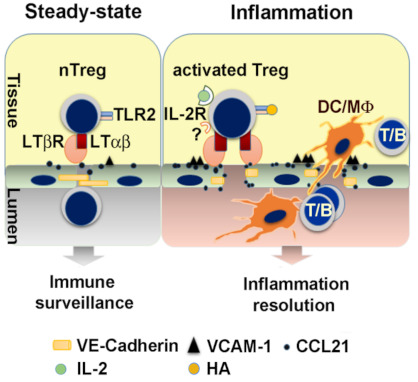
Treg license LEC via LTα1β2-LTβR to facilitate other immune cell lymphatic migration. In homeostatic conditions, patrolling nTreg maintain LEC LTβR constitutive activation, and permit both Treg and naïve or memory T cell recirculation to maintain immune surveillance. In inflammation, activated Treg with the highest LTα1β2 expression trigger LTβR signaling on LEC and increase VCAM-1 and CCL21, and decrease the intercellular tight junction protein VE-cadherin (VE-cad). These changes facilitate the TEM of other immune cells such as dendritic cells (DC), macrophages (Mϕ), B cells, and T cells (including activated CD4, CD8). Activation of TLR2 by endogenous ligands such as hyaluronan (HA) intensifies LEC LTβR signaling (Adopted from [51]).

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
