# Peer review of "LTβR Signaling Controls Lymphatic Migration of Immune Cells"

_cells, 2021, doi:10.3390/cells10040747_

Round 1

Reviewer 1 Report

In this review the authors summarize the current knowledge concerning the surface receptor LTbR and describe its involvement in the regulation of immune cell migration, specifically regarding the Treg subsets. They moreover describe in detail the signaling cascades, both canonical and non-canonical, activated by the interaction of LTbR with congnate ligands, and the regulation of transendothelial migration of both Tregs and other immune cells. They moreover describe the potential therapeutic effects of LTbR targeting on cancer, autoimmune diseases, and transplant rejection.

The review is interesting, and I have no specific concerns regarding the scientific contents. However, I strongly suggest a profound revision of the style since the meaning of some sentences is sometimes not easily understandable. Here and there, the sentences are barely connected each other, and this strongly affects the understandability of the text. Moreover, some sentences need to be rewritten, since they appear unclear. See for example the sentence “In unstimulated cells…”, lines 123-127, whose meaning is not clear to me.

Minor points.

1) Lines 102-115. Please refer to figure 2.

2) Lines 112-113. I suggest cutting from the text, or at least rewriting this sentence since it does not add anything, rather it appears unbound to the rest of the text.

3) Lines 210-212. Please add the reference to the sentence.

4) Line 262. Avoid repeating NF-kB definition. The same for all the other acronyms, which must be explicated only the first time.

Author Response

Thank you for the constructive suggestions which would significantly improve the quality of the manuscript. Please find below a point-by-point response to each of the questions and issues.  We have made many corrections based on all reviewer 1's suggestions .

Review 1

The review is interesting, and I have no specific concerns regarding the scientific contents. However, I strongly suggest a profound revision of the style since the meaning of some sentences is sometimes not easily understandable. Here and there, the sentences are barely connected each other, and this strongly affects the understandability of the text. Moreover, some sentences need to be rewritten, since they appear unclear. See for example the sentence “In unstimulated cells…”, lines 123-127, whose meaning is not clear to me. 

We modified some sentences accordingly to enhance the clarity and readability. Such as “in unstimulated cells” to “in resting cells”.

Minor points. 

1) Lines 102-115. Please refer to figure 2. 

We added “Figure 2” in section 3

2) Lines 112-113. I suggest cutting from the text, or at least rewriting this sentence since it does not add anything, rather it appears unbound to the rest of the text. 

We deleted the sentence, “IKK is comprised of…”

3) Lines 210-212. Please add the reference to the sentence. 

The reference is added now to the sentence, “Recently, our group recognized…”

4) Line 262. Avoid repeating NF-kB definition. The same for all the other acronyms, which must be explicated only the first time. 

We deleted the repeated definition, and carefully checked for other unnecessary repeats.

Reviewer 2 Report

This is a well-organized, highly informative review article on the role of LTbR on LEC for immune cell migration, specifically Treg cells. A few specific suggestions are raised as below.

  1. LEC-based immune cell migration typically includes dendritic cells, in addition to Treg cells. However, no DC migration is discussed in this review. Recently it was reported that skin DC migration to draining lymph nodes is regulated by Langerhans cell-derived LIGHT and LEC-derived LTbR via CCL21/CC19 expression (Wang et al. J Immunol. 2019 May 15;202(10):2999-3007.). As a typical LEC-based immune cell migration, this study should be discussed, probably in section 6.

  1. Even the discussion about DC migration as mentioned above is added, I understand it might be far less comprehensive than that on Treg. The discussion about Treg cells would still be the predominant part of the article. Therefore, it might not be accurate to use “immune cells” in the title. “Treg cells” is suggested to use, unless the discussion about DC migration is greatly expanded.

  1. As to the ligand of LTbR for LEC regulation, only LT was discussed in this article. However, as I mentioned above, LIGHT is obviously involved in LEC activation via LTbR for DC migration. This needs to be clarified, probably also in section 6.

Other than these, this is a pretty good review.

Author Response

Thank you for the constructive suggestions which would significantly improve the quality of the manuscript. Please find below a point-by-point response to each of the questions and issues.  We have made many corrections based on all reviewers’ suggestions and added more about “LIGHT-LTbR signaling on DC migration” to section 6 based on reviewer 2’s suggestion.

Reviewer 2: 

This is a well-organized, highly informative review article on the role of LTbR on LEC for immune cell migration, specifically Treg cells.

A few specific suggestions are raised as below. 

1. LEC-based immune cell migration typically includes dendritic cells, in addition to Treg cells. However, no DC migration is discussed in this review. Recently it was reported that skin DC migration to draining lymph nodes is regulated by Langerhans cell-derived LIGHT and LEC-derived LTbR via CCL21/CC19 expression (Wang et al. J Immunol. 2019 May 15;202(10):2999-3007.). As a typical LEC-based immune cell migration, this study should be discussed, probably in section 6. 

We added more about DC migration regulated by LIGHT-LTbR signaling in section 6.

2. Even the discussion about DC migration as mentioned above is added, I understand it might be far less comprehensive than that on Treg. The discussion about Treg cells would still be the predominant part of the article. Therefore, it might not be accurate to use “immune cells” in the title. “Treg cells” is suggested to use, unless the discussion about DC migration is greatly expanded. 

Yes, the LTbR signaling-regulated Treg migration is the predominant part of this review. Regulation of the migration of other immune cells is related to the effects of LTa1b2-LTbR signaling by Tregs which then conditions LEC to facilitate the migration of other immune cells, including DCs, macrophages, and B cells (see section 8.2).

3. As to the ligand of LTbR for LEC regulation, only LT was discussed in this article. However, as I mentioned above, LIGHT is obviously involved in LEC activation via LTbR for DC migration. This needs to be clarified, probably also in section 6. 

We added LIGHT stimulated LTbR signaling for DC migration in section 6. 

Round 2

Reviewer 2 Report

My previous concerns have been satisfactorily addressed.